# On the alignment of LM language generation and human language comprehension

**Lena S. Bolliger,   Patrick Haller,   Lena A. Jäger**
Department of Computational Linguistics, University of Zurich, Switzerland
{bolliger,haller,jaeger}@cl.uzh.ch

## Abstract

Previous research on the predictive power (PP) of surprisal and entropy has focused on determining which language models (LMs) generate estimates with the highest PP on reading times, and examining for which populations the PP is strongest. In this study, we leverage eye movement data on texts that were generated using a range of decoding strategies with different LMs. We then extract the transition scores that reflect the models' *production* rather than comprehension effort. This allows us to investigate the alignment of LM language production and human language comprehension. Our findings reveal that there are differences in the strength of the alignment between reading behavior and certain LM decoding strategies and that this alignment further reflects different stages of language understanding (early, late, or global processes). Although we find lower PP of transition-based measures compared to surprisal and entropy for most decoding strategies, our results provide valuable insights into which decoding strategies impose less processing effort for readers. Our code is available via https://github.com/DiLi-Lab/LM-human-alignment.

## 1 Introduction

Human language processing is incremental in nature: words are processed sequentially, and each word might require a different amount of cognitive effort to be expended (Rayner, 1998; Rayner and Clifton, 2009) depending on how predictable it is in its current context. This relationship between cognitive processing effort and word predictability is operationalized in Surprisal Theory (Hale, 2001; Levy, 2008), which posits that the cognitive effort is proportional to word predictability, quantified as surprisal, the negative log-probability of a word conditioned on its preceding context. Leveraging reading times (RTs) as proxy for cognitive effort and employing neural language models (LMs) to

estimate surprisal, this relationship has been corroborated extensively (Demberg and Keller, 2008; Goodkind and Bicknell, 2018; Wilcox et al., 2020; Shain, 2021; Pimentel et al., 2021; Kuribayashi et al., 2021; Shain et al., 2024; Wilcox et al., 2023b, *i.a.*). While these studies assume that the reading process is purely responsive, *i.e.*, readers allocate time to process a word as they encounter it, other studies argue that the reading process might additionally be anticipatory in nature (Pimentel et al., 2023): readers make predictions about an upcoming word and, based on this expectation, *preemptively* assign time to its processing, which affects reading behavior. This anticipatory predictability effect is quantified as contextual entropy (Shannon, 1948; Hale, 2006), the expectation over a word's surprisal, and has been found to be predictive of reading times as well (Linzen and Jaeger, 2016; van Schijndel and Schuler, 2017; Wilcox et al., 2023b; Pimentel et al., 2023). Research on surprisal and contextual entropy has relied on the notion of (psychometric) predictive power (PP), which quantifies the fit (*i.e.*, performance) of a regression model on RTs including a predictor of interest (surprisal or contextual entropy) in comparison to a baseline model. These studies on PP have been conducted along several axes. The first tackles the question of which LMs estimate these metrics such that they exhibit the highest PP on RTs, investigating LM family (Shain et al., 2024), LM quality (Wilcox et al., 2020, 2023b), LM size (Oh and Schuler, 2023b), and the amount of training data (Oh and Schuler, 2023a). Another axis involves shifting the focus on the population whose RTs are predicted, such as speakers of different languages (Wilcox et al., 2023b) or groups of readers representing different cognitive profiles (Škrjanec et al., 2023; Haller et al., 2024).

While these studies on the PP of surprisal and entropy have explored the alignment between LM *comprehension effort* and human reading behavior,

we introduce a third axis of investigating PP that *directly assesses the alignment of LM language pro-duction and human language comprehension*. We shift the focus from the LMs that estimate the pre-dictability metrics directly to the texts being read. To that end, we leverage the Eye Movements on Machine-Generated Texts Corpus (EMTeC; Bol-liger et al., 2024) that contains reading data on En-glish texts generated with different large language models (LLMs) and different decoding algorithms, and that further provides the LLMs' raw generation transition scores. This allows for investigating what role the nature of the text itself plays for human reading behavior. More specifically, it enables us to i) disentangle the alignment of human language processing with certain LMs and certain decoding strategies, and ii) to assess whether information about the text generation process improves the PP of surprisal and contextual entropy on these texts. Typical language generators define a probability distribution over sequences of tokens, which can be understood as the model's uncertainty about gen-eration given a context (Giulianelli et al., 2023a). With humans experiencing both responsive as well as anticipatory effects in reading, we assume there exists an alignment between LMs and humans in that LMs' uncertainty during language production is reflected in the uncertainty humans experience during language comprehension.

After conducting a baseline analysis ($RQ_B$) that establishes the PP of surprisal and contextual en-tropy on the EMTeC stimuli, where we estimate the predictability metrics both with GPT-2 *base* (Rad-ford et al., 2019) as well as with the LLMs used to generate the stimuli, we investigate the following research questions:

$RQ_1$ To what extent do different decoding strate-gies and human language comprehension align, and does this alignment reflect respon-sive or anticipatory processing?

$RQ_2$ Which decoding strategies generate texts that elicit low (or high, respectively) surprisal and entropy effects in humans?

$RQ_3$ Do surprisal and contextual entropy ex-tracted from the stimuli's transition scores exhibit greater PP than surprisal and contex-tual entropy estimated with neural language models?

We fit our models on a variety of reading measures (RMs) that include both binary as well as continu-ous measures which can be divided into measures of early, late, and global language processing. Our findings suggest that certain decoding strategies align better with human language processing than others and underline the notion of selecting LMs and reading measures based on the specific cogni-tive processes under investigation, such as early or late reading processes.

## 2 Related Work

At present, relatively little is understood as to whether LMs and humans process texts in a similar way. Giulianelli et al. (2023a) evaluated LMs in terms of whether their representation of uncertainty is calibrated to the levels of variability observed in humans by comparing LMs' distributions over productions against the distributions over the pro-ductions of humans, given the same context. They found that LMs capture human variability well (though not as well as another human) with most decoding algorithms, though ancestral sampling matched the plausible space of human productions closest. Similarly, Venkatraman et al. (2023) inves-tigated whether decoding algorithms implicitly fol-low the UID (Uniform Information Density) prin-ciple, which states that humans distribute informa-tion in their utterances evenly. They generated texts with greedy search and ancestral, top-$k$, and top-$p$ sampling and collected human judgments, and found non-uniformity to be a more desirable prop-erty in machine-generated texts, with UID scores not correlating with human judgments. In another study, Giulianelli et al. (2023b) present *information value*, a metric quantifying the predictability of an utterance relative to a set of alternatives. They ob-served that information value has higher PP than aggregates of token-level surprisal for acceptability judgments, and is on par with aggregated surprisal as a predictor of RTs. They further state that the decoding strategies used to generate the utterances do not impact the PP. And last, Liu et al. (2024) in-vestigated what effect temperature-scaling of LLM predictions has on surprisal estimates and demon-strated that temperature-scaled surprisal (with a temperature $T \simeq 2.5$) improves PP on RTs. This underlines their assumption that human probability distributions might be flatter than those learned by LMs. The studies investigating the effect of de-coding algorithms (Giulianelli et al., 2023a; Venka-traman et al., 2023) did not employ human cogni-tive data, while Giulianelli et al. (2023b) explored sentence-aggregates. Our study is a departure from

their approaches in that it leverages cognitive data on machine-generated texts and can thus directly investigate LM and human alignment.

## 3 Methods

In the following, let $w_t$ be word $w$ at index $t$, and let $\boldsymbol{w}_{<t}$ be the sequence of words preceding $w_t$, *i.e.*, its left context. Let $\Sigma$ denote the vocabulary, and $\bar{\Sigma} = \Sigma \cup \{\texttt{EOS}\}$ an augmented vocabulary containing a special EOS (end-of-sentence) token.

### 3.1 Surprisal

The information contained by a word $w_t$ has been quantified by Shannon (1948) as that word's negative log-probability given its preceding context. This quantity was later formalized as surprisal (Hale, 2001; Levy, 2008), and the surprisal $s$ of a word is defined as

$$s(w_t) := -\log_2 p(w_t \mid \boldsymbol{w}_{<t}),$$

where $p(\cdot \mid \boldsymbol{w}_{<t})$ is the true distribution over words $w \in \bar{\Sigma}$ in context $\boldsymbol{w}_{<t}$. This distribution, however, is unknown, and surprisal is commonly estimated by an autoregressive language model $p_\phi$, *i.e.*, $s(w_t) \approx -\log_2 p_\phi(w_t \mid \boldsymbol{w}_{<t})$.

### 3.2 Contextual entropy

The contextual entropy $H$ of a $\bar{\Sigma}$-valued random variable $W_t$ at index $t$ is the expected value of its surprisal, formalized as

$$H(W_t \mid \boldsymbol{W}_{<t} = \boldsymbol{w}_t) := \mathbb{E}_{w \sim p(\cdot \mid \boldsymbol{w}_{<t})} [s_t(w)]$$
$$- \sum_{w \in \bar{\Sigma}} p(w \mid \boldsymbol{w}_{<t}) \log_2 p(w \mid \boldsymbol{w}_{<t}).$$

It is a specific form of the Shannon entropy $H(W) := -\sum_{w \in W} p(w) \log p(w)$ (Shannon, 1948) that is conditioned on the left context of $W$. Again, we do not have access to the true distribution $p(\cdot \mid \boldsymbol{w}_{<t})$ and approximate it with $p_\phi$.

### 3.3 Psychometric Predictive Power

The predictive power of surprisal or entropy refers to the extent of their capacity to predict reading times (RTs). One commonly used approach is to utilize generalized linear-mixed models.[1] Let $\mathcal{M}_\theta : \mathbb{R}^d \to \mathbb{R}$ be a linear-mixed model parametrized by $\theta$, mapping from $d$ predictors to a log-transformed reading time measure $y_{ij}$ obtained from subject $j$

on word $i$, and let $\boldsymbol{v}_i = (v_{1i}, \ldots, v_{di})^\top \in \mathbb{R}^d$ be a set of predictors assumed to affect RTs, such as lexical frequency $f(w_i)$ and word length $l(w_i)$, of word $i$. Then $\mathcal{M}_\theta : \boldsymbol{v}_i \mapsto y_{ij}$.

To assess the predictive power of a single predictor, we follow previous research (Wilcox et al., 2020; Meister et al., 2021; Wilcox et al., 2023a; Pimentel et al., 2023; Haller et al., 2024) in operationalizing predictive power as the mean difference in log-likelihood $\Delta_{\text{LL}}$ between a baseline regression $\mathcal{M}_\theta^b : \boldsymbol{v}_i^b \mapsto y_{ij}$, where $\boldsymbol{v}_i^b$ contains baseline predictors, and a target regression $\mathcal{M}_\theta^t : \boldsymbol{v}_i^t \mapsto y_{ij}$, where $\boldsymbol{v}_i^t$ contains both the baseline predictors as well as a target predictor (predictor of interest). Then $\Delta_{\text{LL}}$ is formalized as

$$\Delta_{\text{LL}} = \frac{1}{IJ} \left[ \sum_{i=1}^{I} \sum_{j=1}^{J} \log \mathcal{M}_\theta^t(y_{ij} \mid \boldsymbol{v}_i^t) \right.$$
$$\left. - \sum_{i=1}^{I} \sum_{j=1}^{J} \log \mathcal{M}_\theta^b(y_{ij} \mid \boldsymbol{v}_i^b) \right],$$

where $I$ is the number of words and $J$ is the number of subjects. To avoid overfitting, we perform 10-fold cross-validation. A positive $\Delta_{\text{LL}}$ indicates that the target predictor increases the predictive power. We fit all models using the R-libraries jglmm (Braginsky, 2024) or lme4 (Bates et al., 2015). To assess statistical significance of $\Delta_{\text{LL}}$, we perform a paired permutation test.

## 4 Experiments[2]

### 4.1 Data

***EMTeC.*** We employ reading data from EMTeC (Bolliger et al., 2024), an English eye-tracking-while-reading corpus of 107 native English subjects whose stimuli were created with the LLMs Phi-2 (Javaheripi et al., 2023) (2.7 billion parameters), the instruction-tuned version of Mistral 7B (Jiang et al., 2023) (7 billion parameters), and WizardLM (Xu et al., 2023) (13 billion parameters), an instruction-tuned version of Llama 2 13B (Touvron et al., 2023). Each model generated texts with different decoding strategies: the likelihood-maximization strategies greedy search and beam search, and the stochastic methods (ancestral) sampling, top-$k$ sampling (Fan et al., 2018), and top-$p$ sampling (Holtzman et al., 2020). With each combination of model and

---

[1] Linear regression on a continuous variable, logistic regression on a binary variable.

[2] Our code is available via https://github.com/DiLi-Lab/LM-human-alignment.

decoding strategy, 42 different texts of maximally 150 tokens were generated, resulting in 588 unique stimuli,[3] and the stimuli belong to six different text types: non-fiction (argumentation or description), fiction (story or dialogue), poetry, text summarization, article synopsis, and key-word based text. EMTeC further provides the raw transition scores of the LLMs' text generation process (*i.e.*, the unnormalized output logits), which compose a distribution over the entire vocabulary at each generation step.

***Reading Measures.*** For our analyses, we consider the continuous reading measures (RMs) *first-pass reading time* (FPRT; the sum of the durations of all first-pass fixations on a word), *re-reading time* (RRT; the sum of the durations of all fixations on a word that do not belong to the first-pass), *total fixation time* (TFT; the sum of all fixations on a word), and *inclusive regression-path duration* (RPD_inc; the sum of all fixation durations starting from the first first-pass fixation on a word until fixating a word to the right of this word including all regressive fixations on previous words),[4] and the binary measures *fixation* (Fix; whether or not a word is fixated) and *first-pass regression* (FPReg; whether or not a regression was initiated in the first-pass reading of the word). While FPRT and FPReg indicate early stages of language processing, RRT and RPD_inc are measures of late processing, and TFT and Fix expresses global processing.[5]

## 4.2 Predictors

We estimate surprisal and entropy with different LMs: GPT-2 *base*, Phi-2, Mistral 7B *base* and *instruct*, and WizardLM. With each LM, we estimate the predictability metrics for the words in the stimuli texts in two ways: first on the stimulus in isolation (unconditioned; used for RQ_B, RQ_1, and RQ_2) and second on the concatenation of prompt[6] and stimulus (conditioned; used for RQ_3). The latter serves the purpose of allowing for comparability of surprisal and entropy with the transition scores, which are inherently conditioned on the prompts. We henceforth denote surprisal and entropy of a word by $s(\cdot)$ and $h(\cdot)$ respectively. Furthermore, we compute the predictability metrics from the

transition score distributions over the vocabulary at each generation step of Phi-2, Mistral *instruct*, and WizardLM, which we will henceforth denote t-surprisal $ts(\cdot)$ t-entropy $th(\cdot)$. As the reading measures are on the level of white-space separated words but LMs employ tokenizers that split such words into sub-word tokens (Sennrich et al., 2016; Song et al., 2021), word-level surprisal is computed by summing up the surprisal values of the individual sub-word tokens. Similarly, word-level entropy is obtained by summing up the sub-word token-level entropy values, which is a proxy for the joint entropy of the sub-word tokens' distributions.[7] We further include the predictors lexical frequency, henceforth denoted $f(\cdot)$, and word length, denoted by $l(\cdot)$. Another predictor is the categorical factor decoding strategy, denoted $dec$, with the levels *beam search, greedy search, (ancestral) sampling, top-k sampling, top-p sampling*.

To avoid terminological confusion, we denote the models Phi-2, Mistral and WizardLM as *surprisal estimation models* when they are used to estimate both surprisal and entropy, and we refer to them as *text generation models* when talking about the stimuli texts the regression models are fitted on with regards to which LLM generated them.

## 4.3 Baseline analysis (RQ_B)

To corroborate previous results on the predictive power of surprisal and entropy, disregarding the effect of decoding strategies and transition scores, all three models used in EMTeC as well as GPT-2 are used as surprisal estimation models to estimate surprisal and entropy. We define a baseline model $\mathcal{M}_\theta^b : \boldsymbol{v}_i^b \mapsto y_{ij}$ with word-level predictors word length $l(w_i)$ and lexical frequency $f(w_i)$, global intercept $\beta_0$, and a random by-subject intercept $\beta_{0j}$:

$$\mathcal{M}_\theta^b : y_{ij} \sim \beta_0 + \beta_{0j} + \beta_1 l_i + \beta_2 f_i,$$

where $y_{ij}$ refers to the log-transformed first-pass reading time (FPRT) of subject $j$ for the $i^{\text{th}}$ word in the stimulus corpus, following a log-normal distribution. The target model $\mathcal{M}_\theta^t : \boldsymbol{v}_i^t \mapsto y_{ij}$ contains as additional predictor either surprisal $s(w_i)$, entropy $h(w_i)$, or both. The regression models are fitted on the entire EMTeC dataset.

***Results.*** As depicted in Figure 1, both surprisal and entropy exhibit significant PP, albeit lower for

---

[3]For Phi-2 beam search, no texts were generated.

[4]Sometimes referred to as "go-past time".

[5]While early measures always indicate early processing, late measures do not exclusively indicate late processes, as early effects might also elicit delayed eye movements.

[6]The prompts used to generate the stimuli in EMTeC.

[7]For details on the pooling of surprisal and entropy, refer to Appendix A.

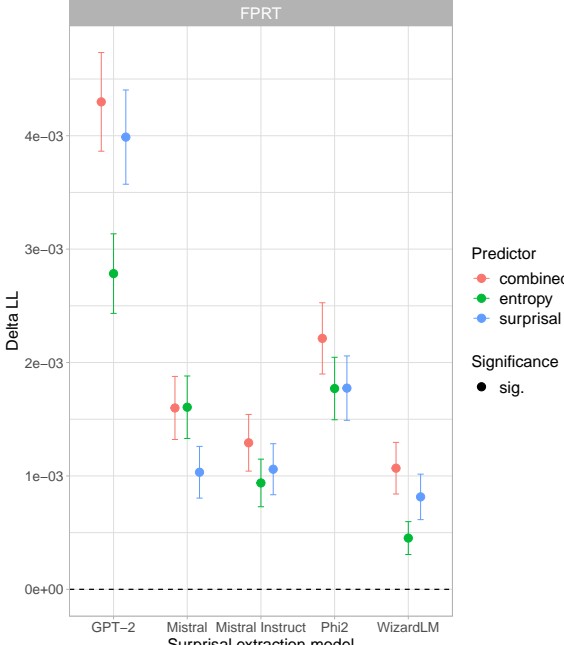

Figure 1: Predictive power of entropy and surprisal on first-pass reading times (FPRT) measured in $\Delta_{LL}$ (mean difference in log-likelihood). *Combined* refers to the regression model in which both predictors were included. Higher $\Delta_{LL}$ indicates higher predictive power. Regression models are fitted on the entire EMTeC dataset.

entropy when estimated with GPT-2 *base*, Mistral *instruct*, and WizwardLM, and lower for surprisal when estimated with Mistral *base*. Adding both surprisal and entropy as predictors improves over using either alone in all cases except for Mistral *base*. Moreover, estimates from GPT-2 have the highest PP, followed by Phi-2, Mistral *base* and *instruct*, with those extracted from WizardLM having the lowest PP. However, the PP of these predictability metrics depends on the predicted reading measure (for the PP on other reading measures, see Figure 5 in Appendix B).

### 4.4 Experiment 1 (RQ$_1$)

We examine the alignment between reading behavior and decoding strategies, *i.e.*, for which texts (generated by a specific combination of LM and decoding strategy) the PP of the predictability metric is the highest. We estimate the metrics with GPT-2 *base*, as it allows for a fair comparison across texts generated with different LMs and has been shown to yield the highest PP (cf. Figure 1). For surprisal as target predictor, we define a baseline model $\mathcal{M}_\theta^b : \boldsymbol{v}_i^b \mapsto y_{ij}$ with word-level predictors $l_i, f_i, h_i$, and by-subject random intercept $\beta_{0j}$ for subject $j$ and the target model $\mathcal{M}_\theta^t : \boldsymbol{v}_i^t \mapsto y_{ij}$

including the predictor of interest $s_i$ such that

$$\mathcal{M}_\theta^b : y_{ij} \sim \beta_0 + \beta_{0j} + \beta_1 l_i + \beta_2 f_i$$
$$\mathcal{M}_\theta^t : y_{ij} \sim \beta_0 + \beta_{0j} + \beta_1 l_i + \beta_2 f_i + \beta_3 s_i.$$

Conversely, for entropy as predictor of interest, the target model includes $h_i$ instead $s_i$. We fit the models separately on the data of each combination of LLM and decoding strategy in EMTeC on the RMs outlined in § 4.1 and compute the $\Delta_{LL}$.

***Results.*** As illustrated in Figure 2, both entropy and surprisal mostly lead to an increase in PP across all LLMs and decoding strategies except when fitted on FPReg. Regarding entropy, there is a strong alignment between top-$p$ and reading patterns for all three LLMs when fitted on RRT and TFT, except for WizardLM with a better alignment with ancestral sampling fitted on RRT. The other RMs do not elicit such a clear pattern. Interestingly, within one LLM, the strength of alignment between decoding strategy and reading behavior differs with respect to the dependent variable (the RM): for instance, for Phi-2, the alignment between FPRT and both ancestral and top-$k$ sampling is greater than with top-$p$ sampling, while for RRT and TFT the pattern is reversed. A similar picture can be observed for surprisal: considering Mistral, for instance, the alignment between ancestral sampling and both TFT and RRT is high, while it is low with FPRT and RPD_inc. For Phi-2, there is an alignment between top-$k$ and FPRT, while for the other RMs the alignment with top-$k$ is weaker than with the other decoding strategies. The alignment of top-$p$ sampling is again high across most reading measures for WizardLM. Again, for the binary RMs Fix and FPReg, no clear alignment pattern is discernible.

### 4.5 Experiment 2 (RQ$_2$)

While the previous experiment investigated which combination of LLM and decoding strategy maximizes the predictive power of surprisal and entropy, here, we adopt the reader perspective and investigate whether the decoding strategy $dec$ a text was generated with impacts the extent to which readers experience a surprisal or an entropy effect. Surprisal and entropy are estimated with GPT-2 *base* for comparative purposes. To do so, we fit a target model $\mathcal{M}_\theta^t : \boldsymbol{v}_i^t \mapsto y_{ij}$ with predictors $l_i, f_i, s_i, dec_i$, an interaction $s_i \times dec_i$, and by-subject random slope $\beta_{0j}$ of subject $j$ as

$$\mathcal{M}_\theta^t : y_{ij} \sim \beta_0 + \beta_{0j} + \beta_1 l_i + \beta_2 f_i + \beta_3 s_i + \\ \beta_4 dec_i + \beta_5 (s_i \times dec_i),$$

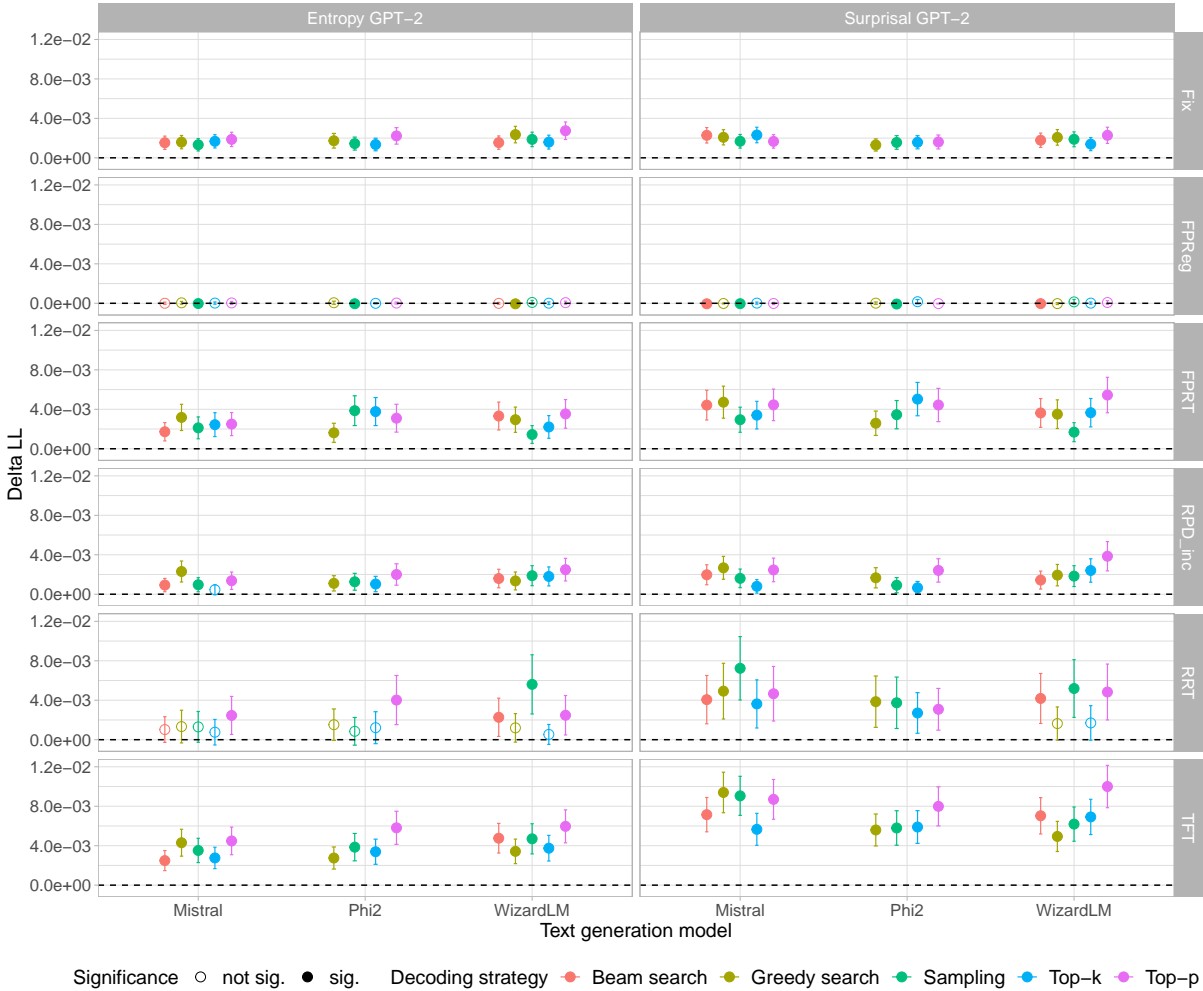

Figure 2: Predictive power (mean and 95% CI) of GPT-2 *base* surprisal and entropy on the prediction of different reading measures measured in $\Delta_{\text{LL}}$. Empty dots indicate that the $\Delta_{\text{LL}}$ is not significantly different from zero. Models are fitted separately on the data of each combination of LLM and decoding algorithm in EMTeC.

where $dec$ is coded via sum contrasts.[8] For the investigation of entropy, we replace $s_i$ with $h_i$ and $(s_i \times dec_i)$ with $(h_i \times dec_i)$. We fit the models separately on the data from each text generation model.

***Results.*** As displayed in Figure 3, there is great variation with respect to the magnitude as well as the significance of the interaction term effects. For WizardLM, the entropy effect reflected in RRT on texts generated with beam search, ancestral sampling, and top-$p$ sampling is significantly different from the grand mean, and the surprisal effect in RRT is significantly greater than the grand mean if the stimuli are generated with beam search and ancestral sampling. Concerning Phi-2, readers ex-

perience a greater-than-average surprisal effect reflected in RPD_inc on texts generated with top-$p$ sampling and reflected in FPReg with greedy search. The entropy effect is above-average in the late(r) measures RRT and TFT on top-$p$ texts, and in FPRT on greedy search texts. The highest number of significant effects are produced by Mistral texts. For instance, texts produced by greedy search, ancestral, and top-$p$ sampling as measured with TFT exhibit a significant entropy effect, as well as beam search, greedy search, and top-$p$ texts reflected in both FPReg and RPD_inc. These results suggest that texts generated by Mistral impose higher processing loads on readers regardless of decoding strategy.

### 4.6 Experiment 3 (RQ₃)

We analyze whether incorporating t-surprisal and t-entropy, *i.e.*, computed from the text generation LLMs' transition scores during stimulus genera-

---

[8]Comparisons consist of decoding strategy minus grand mean (average across all decoding strategies). For the contrast matrix to be singular, comparison with one decoding strategy must be dropped (top-$k$ sampling for Mistral and WizardLM, beam search for Phi-2.)

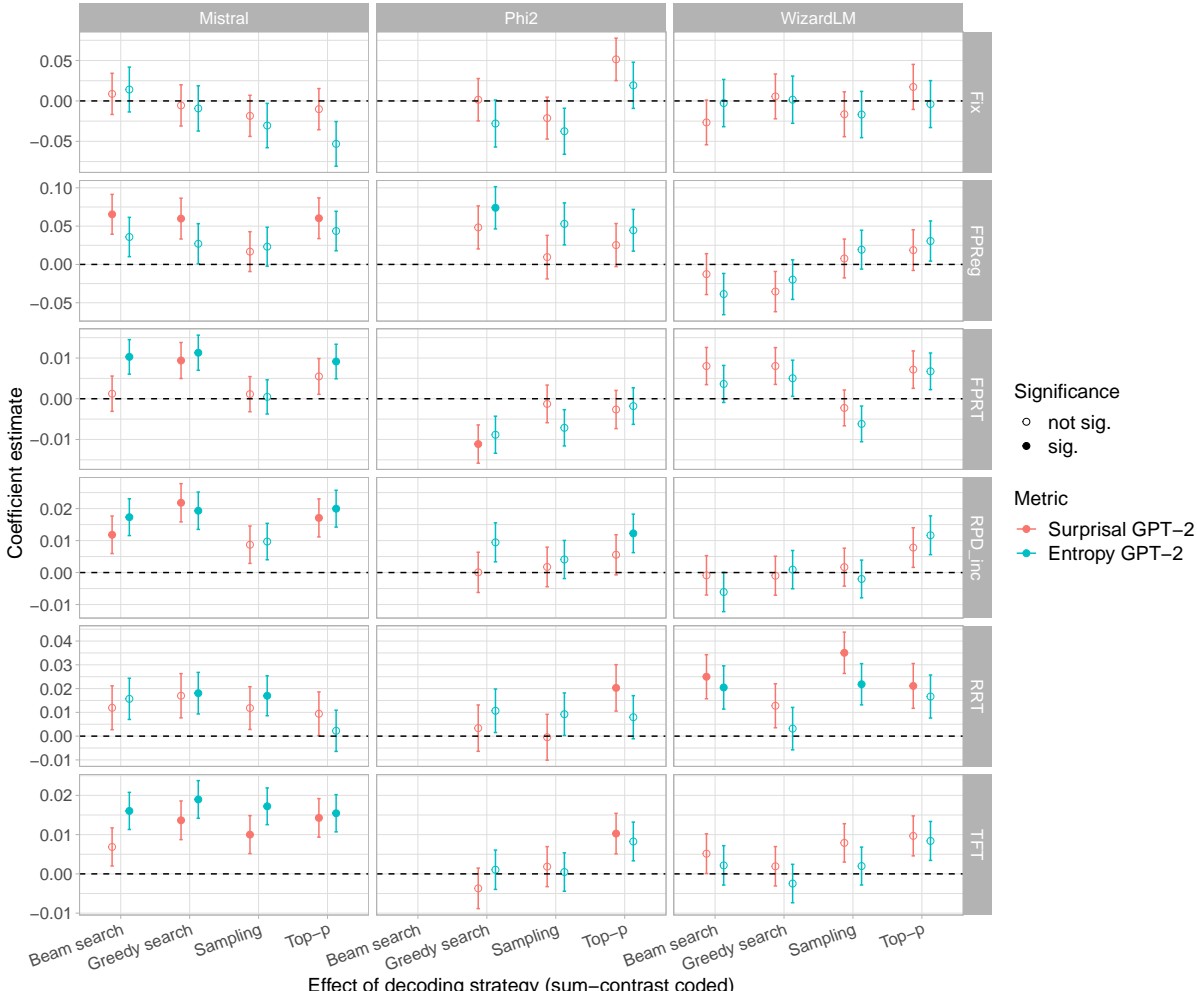

Figure 3: Effect sizes (mean and 95% CI) of the interaction terms between contrast-coded decoding strategy (sum contrasts; decoding strategy minus grand mean) and GPT-2 *base* surprisal (red) and entropy (blue) across text generation models and reading measures. A filled circle indicates that the interaction is statistically significant. Models are fitted separately on the data generated by the different LLMs in EMTeC.

tion, lead to an increased PP over surprisal and entropy extracted from those same models. To that end, we define a baseline model $\mathcal{M}_\theta^b : \boldsymbol{v}_i^b \mapsto y_{ij}$ and a target model $\mathcal{M}_\theta^t : \boldsymbol{v}_i^t \mapsto y_{ij}$ such that

$$\mathcal{M}_\theta^b : y_{ij} \sim \beta_0 + \beta_{0j} + \beta_1 l_i + \beta_2 f_i + \beta_3 s_i$$
$$\mathcal{M}_\theta^t : y_{ij} \sim \beta_0 + \beta_{0j} + \beta_1 l_i + \beta_2 f_i + \beta_3 t s_i,$$

where $s_i$ and $ts_i$ is replaced with $h_i$ and $th_i$ for the investigation of entropy. $s_i$ and $h_i$ are estimated with the very models used to generate the EMTeC stimuli,[9] and by concatenating stimuli and their prompts, which ensures direct comparability with $ts_i$ and $th_i$. We fit the models separately on combinations of LLM and decoding strategy.

***Results.*** The results are displayed in Figure 4. We observe a significant increase in PP of t-surprisal

---

[9]For Mistral, we include both the *base* and the *instruct* model.

over surprisal for stimuli generated with WizardLM using beam search and greedy search. Beyond that, there is no significant increase in PP across models and decoding strategies. While for Phi-2, the $\Delta_{\mathrm{LL}}$ with respect to surprisal is not significant, entropy leads to a significant increase in PP over t-entropy for texts generated with ancestral, top-$k$, and top-$p$ sampling. Regarding Mistral, surprisal estimated with the *base* model has significantly increased PP over t-surprisal for beam search, ancestral and top-$k$ stimuli and only for beam search when estimated with the *instruct* model. The same goes for entropy estimated with the *base* model for ancestral and top-$p$ sampling and for top-$k$ and top-$p$ sampling when estimated with Mistral *instruct*).

The results for baseline and target models fitted on TFT, RRT, RPD_inc, Fix, and FPReg are depicted in Appendix C. While the $\Delta_{\mathrm{LL}}$ is still mostly not significantly different from zero or sig-

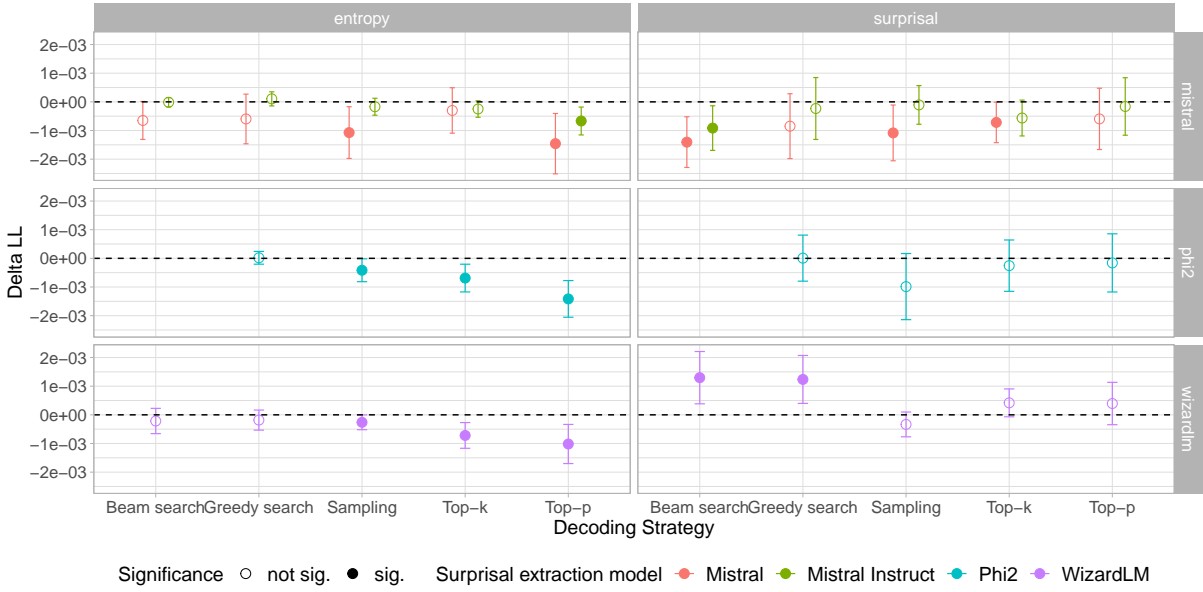

Figure 4: Predictive power (mean and 95% CI) of t-surprisal and t-entropy over surprisal and entropy on FPRT. A triangle indicates that the $\Delta_{\text{LL}}$ is significantly different from zero. A negative $\Delta_{\text{LL}}$ indicates that the baseline has greater predictive power.

nificantly lower than zero, when fitted on certain reading measures, the transition-score based predictability metrics have increased PP for certain combinations of model and decoding strategy (e.g., t-entropy compared to Mistral *instruct* entropy for beam search and compared to Phi-2 entropy for greedy search fitted on Fix; or t-surprisal over both Mistral *base* and *instruct* surprisal for top-$k$ texts fitted on RPD_inc.

## 5 Discussion

The experimental results presented in this study contribute to the understanding of the alignment between language models and human reading behavior. This is particularly evident in the way the texts generated with certain decoding strategies elicit predictability effects that are aligned with reading behavior reflecting either early or late language processing mechanisms. The baseline analysis (see § 4.3) corroborates previous findings stating that different LMs produce predictability estimates with varying predictive power, and that those yielded by GPT-2 *base* generally have the highest PP (Shain et al., 2024). However, this analysis also underscores the notion that the choice of LM as predictability metric estimator depends on the aspect of reading behavior one is interested in researching: some LMs better capture anticipatory reading effects via entropy than responsive effects via surprisal. This is further reinforced and ex-

panded upon in the investigation of RQ$_1$ (see § 4.4), which aimed at investigating the extent to which different decoding algorithms and human language comprehension align. The alignment patterns of the different decoding strategies are not consistent across reading measures: for instance, considering surprisal predicting RMs on Mistral texts, the alignment is high with ancestral sampling for RRT and TFT but low for FPRT, and considering entropy, top-$p$ sampling exhibits high alignment for RRT and TFT across the three LLMs. This suggests that the alignment of a decoding strategy with reading behavior hinges on the RM the regression is fitted on. On the one hand, these different alignment patterns exemplify that different models, combined with different decoding strategies, produce texts that are more or less aligned with human language comprehension. On the other hand, this variability of alignment between RMs also suggests that making a claim for the "best overall fit" of surprisal and entropy might not be sensible. Most previous studies (*i.a.* Wilcox et al., 2023b,a; Pimentel et al., 2023) have focused on FPRT, as it reflects initial processing difficulty and is purportedly most aligned with LM surprisal due to the autoregressive nature of language models. However, we argue for choosing an RM that best approximates human expectation-based reading behavior with respect to a specific reading process one is investigating, as reflected in early, late, or global measures.

Apart from the choice of LM and the implica-

tions of the choice of RM as dependent variable, we also find differences in the alignment between human reading behavior and expectation-based effects observed on texts generated by different decoding algorithms for a variety of RMs, as well as differences in the strength of the interactions between decoding strategy and predictability metrics. This allows for the interpretation of whether the texts generated with certain LMs, and certain decoding strategies in particular, require larger cognitive effort from the reader at different stages of processing. As explored in $RQ_2$ (see § 4.5), where we examined which decoding algorithm generates texts that result in low or high surprisal or entropy effects, Phi-2 texts generated with top-$p$ sampling elicit large surprisal and entropy effects across late and global RMs (RRT, TFT, RPD_inc), while texts generated with greedy search lead to smaller surprisal effects in FPRT. For Mistral-generated texts, ancestral sampling, top-$p$ sampling, and beam search also result in higher effects observed in TFT, RRT and RPD_inc. Pertaining to WizardLM, we find that on texts generated via beam search, ancestral sampling and top-$p$, high surprisal words cause significantly higher RRTs. These results imply that WizardLM and Mistral likely generate texts that disrupt late stages of processing regardless of the decoding strategy. For Phi-2, on the other hand, generating texts using greedy search leads to facilitated early-stage processing.

Whereas there might be support for the claim that stochastic strategies are cognitively more plausible than likelihood-maximization strategies (Holtzman et al., 2020), we refrain from directly linking the mechanisms underlying the stochastic strategies (such as re-distribution for top-$p$) with the cognitive mechanisms in humans. While in the analysis of $RQ_3$ (see § 4.6), we find that t-surprisal improves the PP over surprisal for texts generated with WizardLM combined with beam search and greedy search, there is mostly no increased PP when computing t-surprisal and t-entropy from the stimuli's transition scores directly. This, in conjunction with the results from $RQ_2$, implies that the alignment of certain decoding algorithms with reading behavior is a result of the properties of the texts these algorithms generate, but that there is no direct reflection of the information contained in the LLMs' text generation transition scores in the reading times. It would thus be far-fetched to claim that language models' generative processes are typifying of the cognitive

processes underlying human language comprehension, and vice versa: we cannot extrapolate from LM generation uncertainty, represented by the transition scores, to human processing difficulty. The alignment between decoding strategies and reading behavior as demonstrated in Experiment 1 (§ 4.4) cannot be predicted by the LLMs' transition scores but may instead be founded in linguistic features of the generated texts.

# 6 Conclusion

We show that (1) the alignment between LMs and human reading behavior varies based on the choice of model and the decoding strategy on the one hand, and on the reading measure used as dependent variable on the other hand; however, the extent of this alignment cannot be inferred from the transition scores; and (2) specific combinations of models and decoding strategies used for text generation impose lower or higher cognitive effort at different stages of processing. This suggests that, when resorting to LMs for the estimation of predictability metrics, psycholinguistic researchers should tailor their selection to the specific language processing stage of interest.

# Acknowledgements

We thank Tannon Kew for providing helpful feedback, and the anonymous reviewers for their valuable comments and insightful suggestions. This article is based upon work from COST Action MultiplEYE (CA21131), supported by COST (European Cooperation in Science and Technology), which is funded by the Horizon 2020 Framework Programme of the European Union.

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

## A  Pooling of surprisal and entropy to word level

The word-level surprisal values are already contained within the EMTeC (Bolliger et al., 2024) dataset, where the subword-level surprisal values are added up to obtain word-level surprisal values. Given $k$ subword tokens $w_n, w_{n+1}, \ldots, w_{n+k}$ belonging to the same word token $w$, the word token surprisal of $w$ is computed as

$$
\begin{aligned}
s(w_n, w_{n+1}, \ldots, w_{n+k}) &= -\log p(w_n, w_{n+1}, \ldots, w_{n+k} \mid \boldsymbol{w}_{<n}) \\
&= -\log \Big[ p(w_n \mid \boldsymbol{w}_{<n}) \cdot p(w_{n+1} \mid \boldsymbol{w}_{<n+1}) \cdot \ldots \\
&\quad \cdot p(w_{n+k} \mid \boldsymbol{v}_{<n+k}) \Big] \\
&= -\log p(w_n \mid \boldsymbol{w}_{<n}) - \log p(w_{n+1} \mid \boldsymbol{w}_{<n+1}) \\
&\quad - \ldots - \log p(w_{n+k} \mid \boldsymbol{v}_{<n+k}).
\end{aligned}
$$

This shows that summing up subword-token surprisal values is equivalent to computing the surprisal of the joint probability distribution of the subword tokens.

Regarding entropy, we use the sum of subword-token-level entropy values as proxy for the joint entropy of the subword tokens' distributions. Given $k$ $\bar{\Sigma}$-valued random variables $W_n, W_{n+1}, \ldots, W_{n+k}$ belonging to the same word token, their joint entropy is defined as:

$$
H(W_n, W_{n+1}, \ldots, W_{n+k}) := - \sum_{w_n \in \bar{\Sigma}} \sum_{w_{n+1} \in \bar{\Sigma}} \cdots
$$

$$
\sum_{w_{n+k} \in \bar{\Sigma}} p(w_n, w_{n+1}, \ldots, w_{n+k}) \log_2 \left[ p(w_n, w_{n+1}, \ldots, w_{n+k}) \right].
$$

However, the cardinality of $\bar{\Sigma}$ could be over 50,000, depending on the tokenizer, which makes the computation of the joint entropy computationally unfeasible. Instead, we use the sum of the individual entropies as proxy. This is only a proxy because

$$
H(W_n, W_{n+1}, \ldots, W_{n+k}) \leq H(W_n) + H(W_{n+1}) + \cdots + H(W_{n+k}).
$$

This inequality is an equality if and only if $W_n, W_{n+1}, \ldots, W_{n+k}$ are statistically independent. Since this is not the case here, the sum of the subword-token-level entropies is used as an upper bound.

# B  Baseline Results

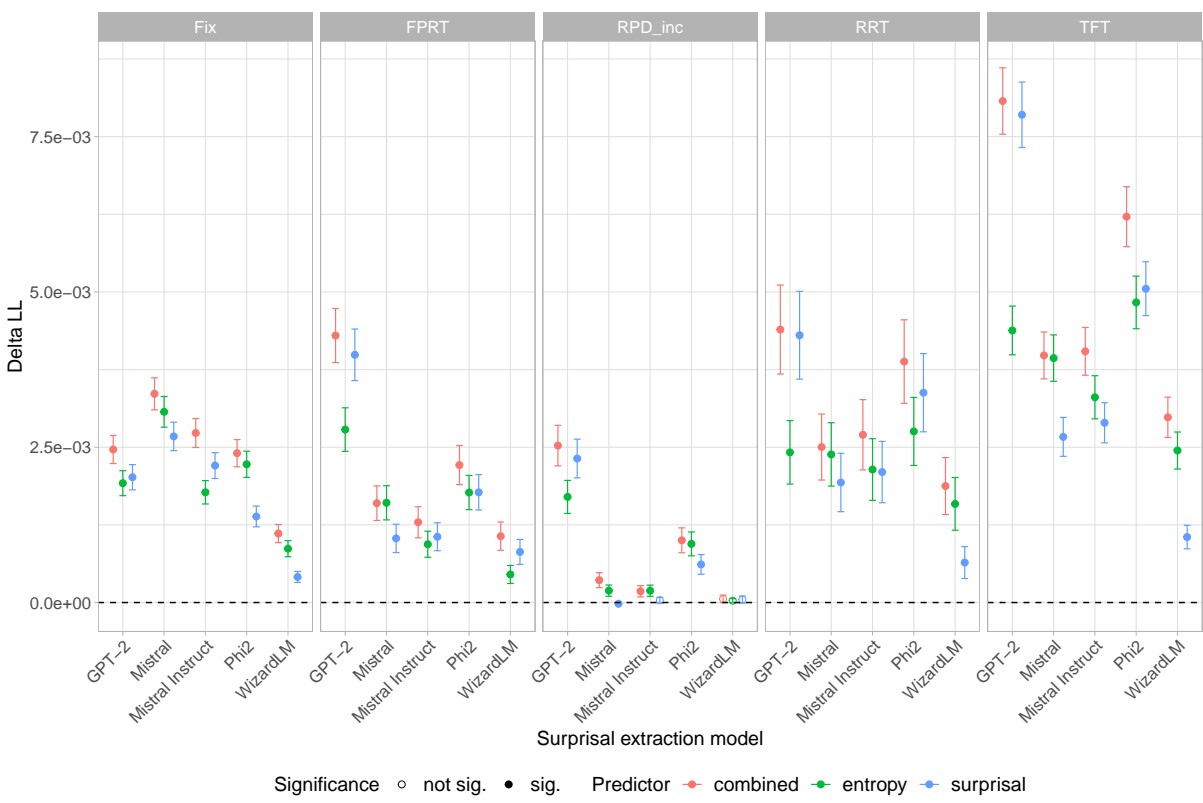

Figure 5: Predictive power of entropy and surprisal on Fix, FPRT, RPD_inc, RRT, and TFT, measured in $\Delta_{\mathrm{LL}}$. Combined refers to the regression model where both predictors are included. Higher $\Delta_{\mathrm{LL}}$ indicates higher predictive power. Regression models are fitted on the entire EMTeC dataset.

# C  Experiment 3 Results

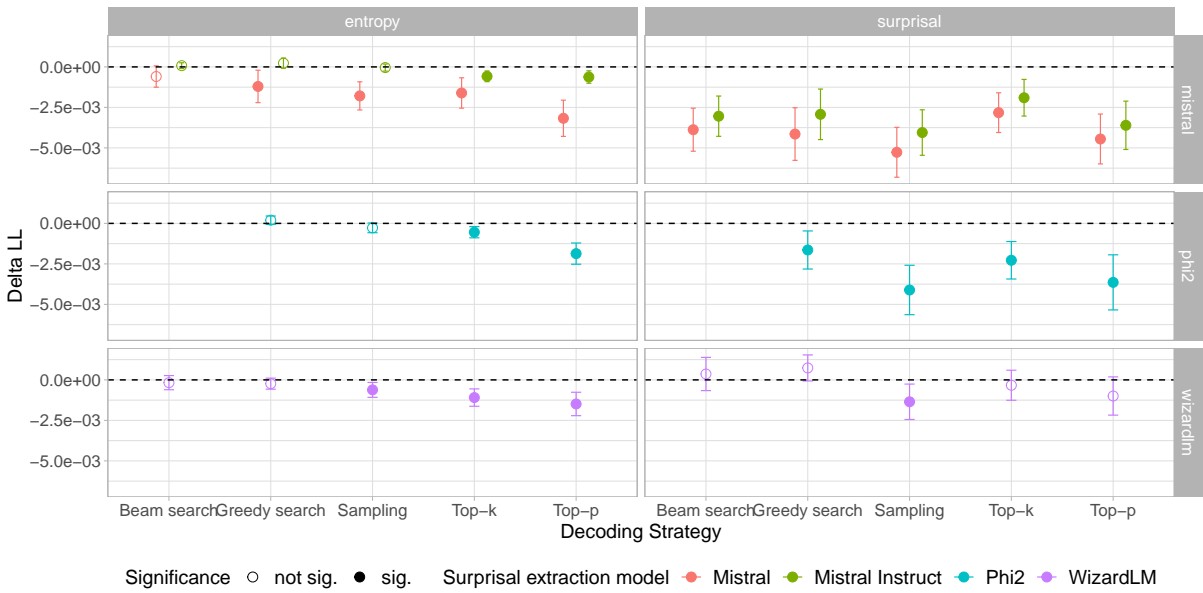

Figure 6: Predictive power (mean and 95% CI) of t-surprisal and t-entropy on TFT. A triangle indicates that the $\Delta_{\mathrm{LL}}$ is significantly different from zero. A negative $\Delta_{\mathrm{LL}}$ indicates that the baseline has greater predictive power.

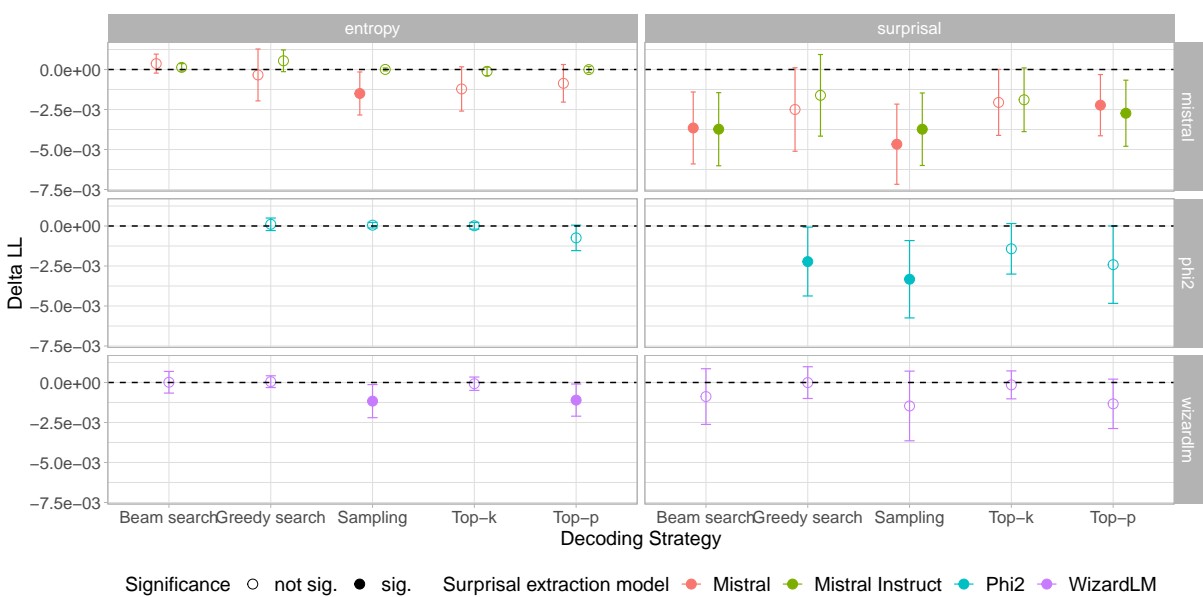

Figure 7: Predictive power (mean and 95% CI) of t-surprisal and t-entropy on RRT. A triangle indicates that the $\Delta_{\mathrm{LL}}$ is significantly different from zero. A negative $\Delta_{\mathrm{LL}}$ indicates that the baseline has greater predictive power.

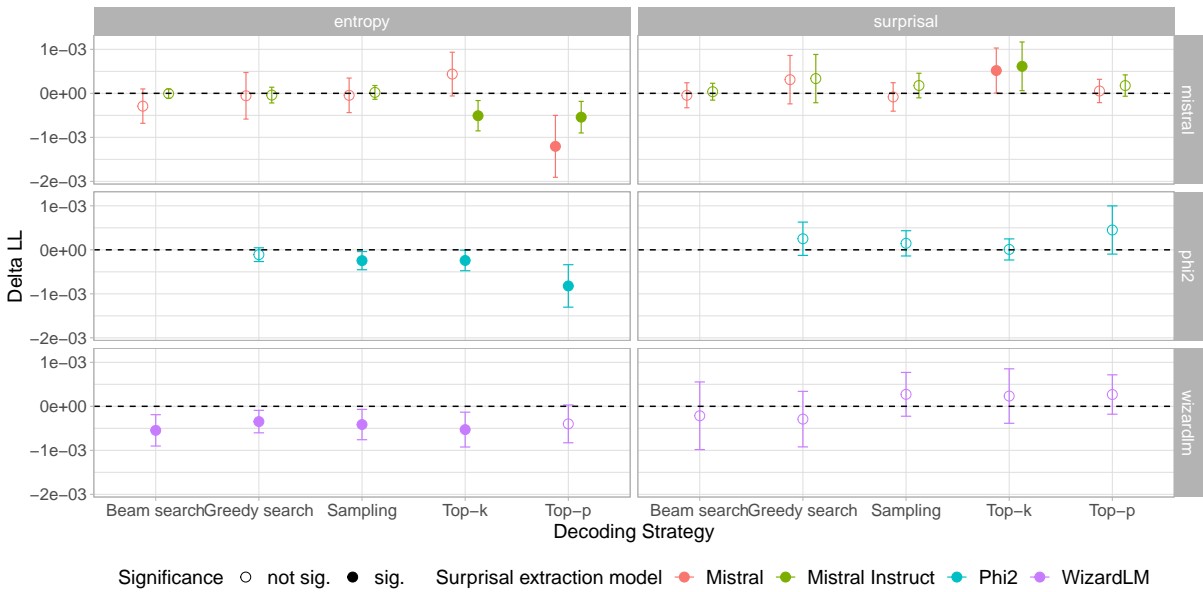

Figure 8: Predictive power (mean and 95% CI) of t-surprisal and t-entropy on RPD_inc. A triangle indicates that the $\Delta_{\mathrm{LL}}$ is significantly different from zero. A negative $\Delta_{\mathrm{LL}}$ indicates that the baseline has greater predictive power.

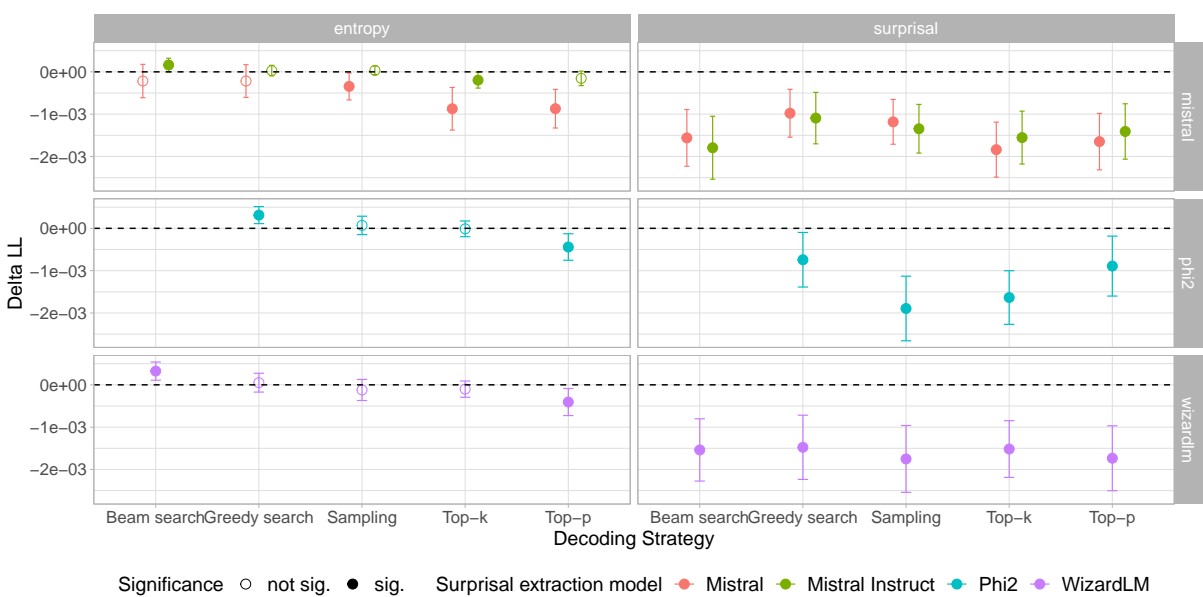

Figure 9: Predictive power (mean and 95% CI) of t-surprisal and t-entropy on Fix. A triangle indicates that the $\Delta_{\mathrm{LL}}$ is significantly different from zero. A negative $\Delta_{\mathrm{LL}}$ indicates that the baseline has greater predictive power.

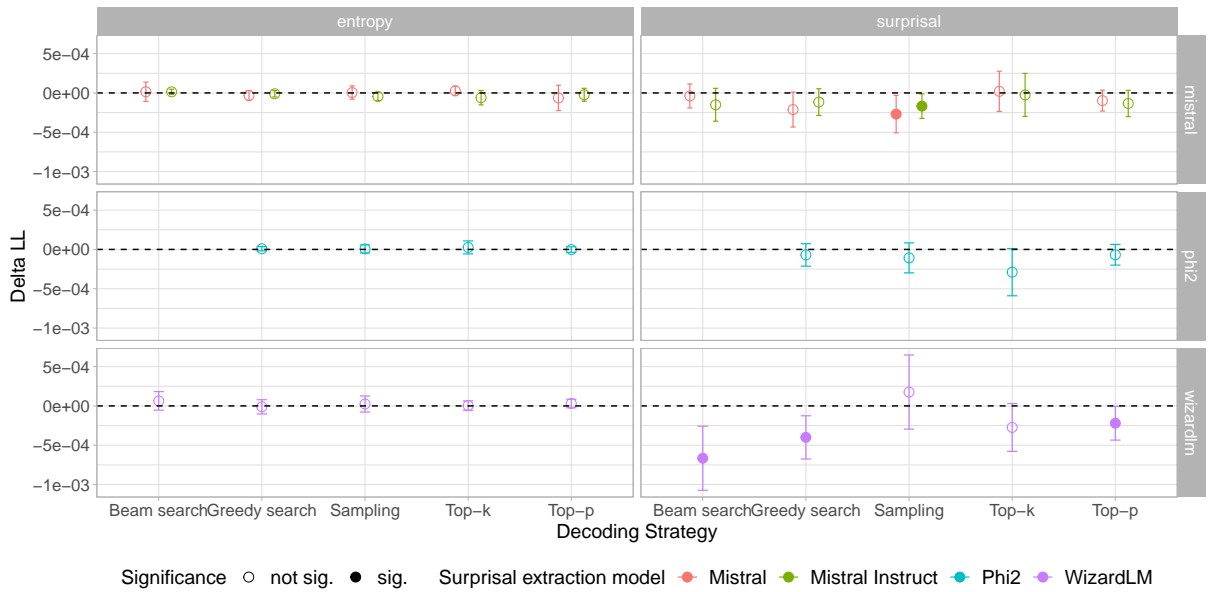

Figure 10: Predictive power (mean and 95% CI) of t-surprisal and t-entropy on FPReg. A triangle indicates that the $\Delta_{\mathrm{LL}}$ is significantly different from zero. A negative $\Delta_{\mathrm{LL}}$ indicates that the baseline has greater predictive power.