# OpenReview forum: "On the alignment of LM language generation and human language comprehension"
_EMNLP/2024/Workshop/BlackBoxNLP — BlackboxNLP 2024_

### Official Review · Reviewer_jUhU · 2024-09-06

**Overall Assessment:** 4
**Confidence:** 2

**Best Paper:**

1

**Best Paper Justification:**

-

**Comments Questions Suggestions And Typos:**

General suggestion:
- For example, I would focus either on the four different language models, or on the decoding strategies, but not on both. Or you could go more in depth on the different levels of human processing.

Figures:
- Figure 2: the texts in grey on top and on the right are barely visible, I noticed them very late. Same for Figures 3 and 4.

**Paper Summary:**

The paper investigates the alignment between generative models and human reading comprehension (through eye movement data), exploring the effect of different decoding strategies on surprisal and entropy, among other things.

**Summary Of Strengths:**

Overall, this looks like a strong paper. It's interesting novel (afaik), the experiments are thorough. The paper is written and structured very clearly. As far as I can understand, the methodology is sound.

**Summary Of Weaknesses:**

Personally, I found the paper a bit dense. The fact of having lots of variables, lots of abbreviations, lots of levels in the analysis can deviate the reader from the main message(s), that should be few(er).  For example, there are multiple reading measures (RMs), the two metrics surprisal & entropy, early vs late vs global processing, different models and different decoding strategies. Also, the paper covers multiple (although related) RQs.

---

### Official Review · Reviewer_aNEy · 2024-09-09

**Overall Assessment:** 3
**Confidence:** 4

**Best Paper:**

1

**Best Paper Justification:**

N/A

**Comments Questions Suggestions And Typos:**

1) All figures and text where applicable: Language models are used in this paper to generate the text stimuli and also calculate surprisal estimates. To avoid confusion (which I had initially), I might recommend explicitly referring to them as "Text Generation Model" (e.g. the x-axis label of Figure 2) and "Surprisal Estimation Model" (e.g. the legend of Figure 4).

2) All figures: I think increasing the font size would greatly improve readability.

3) All figures: Where are the results on text generated using Llama 2?

4) Figure 1: I think this should be moved closer to where this is first referred to in the text.

5) Line 100: tokens , $\rightarrow$ tokens,

6) Line 233: This is a minor point, but the current equation assumes there is data for every subject-by-word combination, which is usually not the case in psycholinguistic modeling. Unless this is actually the case for Bolliger et al.'s (2024) dataset, $\frac{1}{IJ}$ could be presented alternatively as $\frac{1}{N}$ (where $N$ is the total number of data points) and the superscripts could be dropped from the sigmas to look like $\Sigma_{i}\Sigma_{j}$.

7) Lines 399-401: Is this correct? Shouldn't the regression models be fit to data pooled across conditions to get estimates for $dec_{i}$?

**Paper Summary:**

This paper studies the word-by-word reading times of text stimuli that have been generated by LLMs (namely Phi-2, Mistral instruct, WizardLM, and Llama 2) through regression analyses that are typical in psycholinguistic modeling work.

The experiments show that 1) surprisal and entropy estimates from GPT-2 base are predictive of first-pass reading times in line with previous studies (Figure 1), surprisal and entropy estimates from GPT-2 base are generally predictive of most reading measures of machine-generated text across LLMs and decoding strategies (Figure 2), there is a lot of variation in the interaction between surprisal/entropy and decoding strategy (Figure 3), and surprisal/entropy derived from transition scores do not yield a better fit to reading measures compared to surprisal/entropy derived directly (Figure 4).

The authors conclude that this suggests different LLMs and decoding strategies influence the reading behavior of human subjects, as well as LM surprisal/entropy's ability to capture it.

**Summary Of Strengths:**

1) Studying the reading times of machine-generated text stimuli is interesting and has potential to shed light on the properties of text generated by LLMs.
2) The regression modeling protocols are solid, and the results are visualized clearly (with room for minor improvements).

**Summary Of Weaknesses:**

1) The biggest concern that I have is whether this work is appropriate for BlackboxNLP. While I think studying the reading times of text generated from LLMs has a lot of potential for studying LLMs as text generation models, but as it currently stands, this paper looks much more like a paper from a cognitive modeling track/venue. I'm also not sure how approachable this paper is to readers that are not familiar with psycholinguistic modeling work.

2) I'm not sure the data supports the overall conclusion that different LLMs and decoding strategies result in differential fit of surprisal/entropy to reading measures. For example, almost all confidence intervals overlap across conditions in Figure 2, which may make readers think that this is just due to chance. I think whether different LLMs and decoding strategies actually result in differential fit of surprisal/entropy should be explicitly tested first, before trying to analyze the trends as is done currently. This could be done by e.g. evaluating whether a "by-LLM" random slope on surprisal/entropy significantly improves the fit of the regression models fit to reading measures pooled across conditions.

3) It seems like there are some important experimental details that are missing, including which will improve the clarity of writing. I'd like to see a more detailed description of Bolliger et al.'s (2024) dataset, including information like how many passages were generated, how long they are, what their 'genres' are, how many subjects were recruited, how many observations were collected, etc. The number of observations collected is also tied to the second concern above about the strength of evidence the data provides.

4) Related to the third concern, how are the transition scores defined? Are they just the raw probabilities cf. the log-transformed surprisal?

---

### Official Review · Reviewer_i8HV · 2024-09-09

**Overall Assessment:** 3
**Confidence:** 3

**Best Paper:**

1

**Best Paper Justification:**

NA

**Comments Questions Suggestions And Typos:**

1. Line 389-392 is difficult to parse.
2. Line 539-541 makes a strong claim without any citation

**Paper Summary:**

This paper investigates the predictive power of surprisal and entropy on eye movement data of reading texts generated using different decoding strategies with different LLMs. The findings reveal that certain decoding strategies align better with human reading processes, particularly in different stages of comprehension. Transition scores do not significantly improve predictive power over traditional surprisal and entropy metrics. Overall, I think this paper would contribute to the discussion about human-LM alignment in the workshop.

**Summary Of Strengths:**

1. evaluated on novel dataset: This study presents a novel approach of assessing human-LM alignment by evaluating the predictive power of conventional metrics on LM generated texts, rather than psycholinguistic stimuli written by humans. By carefully comparing the performance on texts generated by different decoding mechanisms, this study contributes to a better understanding of the effect of decoding strategies and different language models.

2. The study evaluates various types of eye movements, offering insights into mechanisms in different stages of human comprehension.

**Summary Of Weaknesses:**

1. I wonder why the first experiment is conducted using GPT-2, Phi-2 and Mistral, whereas the dataset is generated with Phi-2, Mistral and WizardLM. Relatedly, does surprisal estimated from a model (e.g. Phi-2) has a better predictive power on text generated from that specific model (e.g. Phi-2) over texts generated from other models?

2. What multiple comparison test is performed in all stats analysis?

3. The paper claims that different decoding strategies impose different processing efforts for readers. I wonder what statistical analysis has been performed. It would be helpful to be clear in the text, about which models have been compared, and the corresponding statistics.

---

### Decision · Program_Chairs · 2024-09-17

**Decision:**

Accept

**Comment:**

Reviewers agree that the results presented in this paper are interesting, novel, and thorough. All reviewers pointed out presentational details that could be improved and certain experimental decisions that should be justified, and I encourage the authors to take these comments into account when preparing the camera-ready.

Reviewer aNEy has concerns on relevance. I think this is a relevant contribution: The stated goal of BlackboxNLP is to "bring together researchers focused on interpreting and explaining NLP models by taking inspiration from fields such as machine learning, psychology, linguistics, and neuroscience." I think this paper fits this theme; moreover, the findings of this paper can shed light on properties of LM behaviors and outputs.